# Investigating the Impact of Ferric Derisomaltose (FDI) on Patient-Reported Quality-of-Life Outcome Measures in Iron-Deficient but Not Anaemic Patients with Chronic Kidney Disease

**DOI:** 10.3390/biomedicines13081860

**Published:** 2025-07-31

**Authors:** Alisha Jafri, Charlotte Youlden, Sebastian Spencer, Sunil Bhandari

**Affiliations:** 1King’s College Hospital NHS Foundation Trust, London SE5 9RS, UK; 2North Cumbria Integrated Care NHS Foundation Trust, Carlisle CA2 7HY, UK; charlotte.youlden2@nhs.net; 3School of Medical Sciences, University of Hull, Hull HU6 7RX, UK; sebastian.spencer2@nhs.net; 4Faculty of Medicine, Hull York Medical School, Hull HU6 7RX, UK; sunil.bhandari@nhs.net; 5Hull University Teaching Hospitals NHS Trust, Hull HU3 2JZ, UK

**Keywords:** chronic kidney disease, iron deficiency, ferric derisomaltose, quality of life

## Abstract

**Background/Objectives**: Iron deficiency without anaemia (IDNA) is common in non-dialysis-dependent chronic kidney disease (CKD) and contributes to fatigue, reduced exercise tolerance, and impaired quality of life (QoL). While intravenous (IV) iron replacement is known to benefit anaemic patients, its role in IDNA remains uncertain. This study aimed to evaluate the impact of ferric derisomaltose (FDI) on patient-reported QoL outcomes in CKD patients with IDNA. **Methods:** This was a post hoc analysis of the double-blind, multicentre Iron and the Heart randomised controlled trial. Fifty-four participants with IDNA (ferritin < 100 µg/L or transferrin saturation < 20% and haemoglobin 110–150 g/L) and CKD stages G3b–G5 were randomised 1:1 to receive either 1000 mg FDI (*n* = 26) or placebo (*n* = 28). An additional 10 iron-replete CKD patients served as controls. SF-36v2 QoL surveys were collected at baseline, 1 month, and 3 months. **Results**: SF-36v2 scores declined across all domains, but deterioration was consistently milder in the FDI group. Role physical declined by 3% in the FDI group versus 12% with placebo and 4% in controls. Bodily pain improved by 2.8% with FDI but worsened by 1.5% in the placebo group. Mental health improved by 3.4 points with FDI and declined by 2.7 points in the placebo group, creating a 6.1-point separation. While differences did not reach statistical significance, likely due to small sample size, the consistent trends favour FDI. **Conclusions**: IV iron may attenuate QoL decline in non-dialysis-dependent CKD patients with IDNA. These findings support the need for larger, adequately powered trials to assess patient-centred outcomes in this population.

## 1. Introduction

Chronic kidney disease (CKD) is a progressive condition characterised by a gradual loss of kidney function, burdening an estimated 10–15% of the global population [1]. This progressive decline in kidney function has various consequences for homeostatic balance, contributing to a range of complications such as metabolic derangements, accelerated cardiovascular disease, and diminished functional capacity [1]. Even in its early stages, the implications of CKD can be extensive for the patient; fatigue, reduced exercise tolerance, and impaired quality of life (QoL) [2]. These symptoms are often attributed to anaemia; however, growing evidence now implicates that iron deficiency itself—independent of anaemia—as a main contributor to the significant symptom burden in this population of patients [3].

Iron plays a central role in oxygen transportation, mitochondrial energy production, and cellular metabolism [3]. In CKD, a convergence of chronic inflammation, impaired intestinal iron absorption, hepcidin dysregulation, and increased iron losses lead to iron-deficiency. This frequently precedes the development of overt anaemia [3]. Despite clinically normal haemoglobin levels, patients experience the functional consequences of iron deficiency, including fatigue, poor exercise tolerance, and cognitive changes [2]. Despite being a distinct and increasingly recognised condition, iron deficiency without anaemia (IDNA) remains underdiagnosed, under-investigated, and undertreated—particularly among patients with CKD [3]. 

Accumulating evidence from other specialist fields underscores the clinical importance of treating iron deficiency irrespective of haemoglobin concentration, particularly to alleviate symptoms. In elderly populations, iron deficiency is consistently linked to impaired muscle function and reduced endurance, independent of anaemia [4]. In people with congestive heart failure (CHF), iron deficiency is associated with skeletal muscle dysfunction, even without overt anaemia [5]. A central mechanism underlying these symptoms is impaired mitochondrial function: iron is essential for the proper functioning of the mitochondrial electron transport chain, where it serves as a cofactor in multiple complexes responsible for ATP production [6]. In iron-deficient states, mitochondrial respiration is compromised, resulting in cellular energy deficits that manifest as fatigue, reduced exercise tolerance, and poor physical performance. These features, common to both CHF and CKD, highlight why addressing iron deficiency, even in the absence of anaemia, is crucial to improving functional outcomes and quality of life.

Iron deficiency contributes significantly to fatigue, lethargy, and reduced physical capacity—symptoms that are common in CKD. In heart failure with reduced ejection fraction, multiple randomised trials have shown that IV iron therapy improves functional capacity, reduces fatigue, enhances QoL, and lowers the risk of hospitalisation and death [7]. Despite growing recognition of the link between iron deficiency and symptoms, iron management in CKD has traditionally centred on correcting anaemia. Standard treatment focuses on restoring haemoglobin levels using erythropoiesis-stimulating agents (ESAs) and iron supplementation, either orally or intravenously, with newer alternatives such as HIF prolyl hydroxylase inhibitors (HIF-PHIs) emerging in recent years [8]. While IV iron is routinely used in people with end-stage kidney disease (ESKD) on haemodialysis, its use in non-dialysis-dependent CKD remains inconsistent and less well established [8]. 

The Iron and the Heart Trial was a double-blinded randomised study in non-anaemic CKD patients with iron deficiency investigating whether IV iron therapy could improve exercise capacity [9]. Key secondary outcomes included patient-reported fatigue levels, measures of QoL, and assessments of muscle strength and physical performance. The study was modelled on heart failure trials, powered to detect similar improvements in exercise capacity with IV iron in CKD. Using data from extended follow-up of the Iron and Heart Study, we aim to further evaluate the efficacy of IV iron in non-dialysis CKD patients with IDNA, across all SF-32v2 domains [10], to capture the broader impact of FDI treatment on daily functioning and well-being. 

## 2. Methods

The Iron and the Heart Trial was a multicentre double-blinded randomised study in which non-anaemic non-dialysis-dependent CKD patients with iron deficiency were given 1000 mg of IV iron (FDI), compared to a placebo infusion at baseline (0 months), and assessed at 1 month and 3 months post-infusion, to evaluate the impact of FDI on exercise capacity [9]. Secondary objectives of the trials assessed the impact of FDI treatment on QoL by utilising SF-36v2 Health Survey. This analysis focuses on the QoL parameters answered by the trial participants as part of the SF-36v2 Health Survey [10].

Fifty-four participants with iron deficiency but no anaemia (Group 1) were randomly assigned to either FDI or placebo (1:1 ratio). Twenty-six participants received FDI, and 28 received placebo. An additional 10 participants with CKD, but without anaemia or iron deficiency (Group 2), did not receive any treatment and were followed up at the same time points as Group 1. 

The placebo administered consisted of 100 mL of 0.9% saline presented as a clear solution, whereas the iron solution appeared brown/black in colour. In the experimental group, 1000 mg of ferric derisomaltose (FDI) was diluted in 100 mL of 0.9% normal saline and administered as a single intravenous infusion over 30 min. In the placebo group, 100 mL of 0.9% normal saline was infused over the same time period. All participants were closely monitored during the infusion and for 30 min afterward to identify any haemodynamic changes or adverse effects.

To maintain blinding, a double screen was used, further covered with a bed sheet, effectively concealing the infusion bag from the participant, the attending physician, and the research nurse. The individual responsible for administering the infusion—who was not involved in data collection or analysis—was able to view the infusion fluid for safety reasons. This approach preserved the integrity of the double-blind design [9].

Randomisation was carried out using a computer program. Labels numbered consecutively from 1 to 60 were generated and placed into opaque, double-sealed envelopes. These envelopes were inaccessible to the investigators. The details of the therapy corresponding to each randomisation number were maintained by the pharmacy, ensuring that the assigned intervention matched the appropriate randomisation number [9].

### 2.1. Study Participants

The criteria for participating in the study included adults with established CKD stages G3b-5, not on dialysis with serum ferritin < 100 mcg/L or transferrin saturation < 20%, without anaemia (Table 1). Within this study, normal haemoglobin was defined as 110–150 g/L for male and female participants [11].

### 2.2. Quality-of-Life Metrics

The KDQoL-SF-36 survey is a disease-specific measure of health-related QoL [12]. It measures 8 health domains: physical functioning, role physical, general health, bodily pain, vitality, social functioning, role emotional, and mental health. Physical functioning reflects levels and limitations between the extremes of physical activity including performed vigorous activities, lifting and carrying groceries, climbing the stairs, bending, kneeling and or stopping, walking moderated distances, and performing self-care activities. Role physical assessed if participants had cut down on the amount of time spent on activities, accomplished less, or were limited or performed tasks with more difficulty because of their condition. Bodily pain evaluated the intensity of bodily pain and the extent of pain interference on normal work activities.

General health included a rating of health and addressing the participants’ expectations of their own health. Vitality assessed differences in subjective well-being. Social functioning assessed the impact of physical health and emotional problems on social activities. Role emotional assessed the limitations related to mental health at work and other regular daily activities. Finally, mental health domain assessed anxiety, depression, loss of behavioural/emotional control, and psychological well-being [4]. The 8 subsections are scored from 0 (worst health) to 100 (best health).

To score the SF-36 survey, scales are standardized with a scoring algorithm and by the SF-36v2 scoring software v1 to obtain a score ranging from 0 to 100 [10]. Higher scores indicate better health status, and a mean score of 50 has been articulated as a normative value for all scales. Collected responses were categorical variables that were summarised using number (%) and then normalized and summarised using mean (standard deviations). Physical functioning, role physical, bodily pain, and general health were combined to form the physical component summary. Vitality, social functioning, role emotion, and mental health were combined to create the mental component summary. These components are all summarised in Table 2.

### 2.3. Trial End Points

Trial end points were based on existing evidence of peak clinical benefit of IV iron, particularly ferric derisomaltose, occurring within the first 4–8 weeks, making 1 month relevant for capturing short-term benefits. The Iron and Heart trial assessed exercise capacity at 1 and 3 months following FDI, citing this as the window during which maximal benefit is observed. [9] The FERWON–IDA trial reported significant improvements in Hb from baseline to 4–8 weeks [13]. From a clinical perspective, assessing outcomes at 1 and 3 months aligns with common patient review timelines, offering a realistic window to detect changes without a prolonged follow-up loss. Safety endpoints included death, infections, vascular access thrombosis, hospitalisation for any cause, and hospitalisation for infection, each assessed during the whole study period. Data on serious adverse events were collected prospectively, and events were coded with the use of the Medical Dictionary for Regulatory Activities (MedDRA), version 15.1. Data on non-serious adverse events, other than infection and vascular access thrombosis, were not collected.

### 2.4. Statistical Analysis

Descriptive statistics were reported as mean ± standard deviation (SD). Within-group changes in SF-36 domain scores (baseline, 1 month, and 3 months) were assessed using repeated-measures ANOVA, with contrasts tested for both linear and quadratic trends. This was performed independently for each intervention group (Placebo, FDI, and CKD without iron deficiency). To assess pre-post differences specifically between baseline and 3 months, paired samples *t*-tests were conducted within each group for each domain. Results are reported with 95% confidence intervals for the mean difference. 

A *p*-value of <0.05 was considered to indicate statistical significance. Between-group comparisons at specific timepoints were evaluated using independent samples *t*-tests. Levene’s test was used to confirm homogeneity of variance, with adjustments made where appropriate. No corrections for multiple comparisons were applied, as this study was exploratory in nature. All statistics were completed using SPSS software: IBM SPSS Statistics (Version 27, Armonk, NY, USA).

## 3. Results

### 3.1. Baseline Characteristics

A total of 54 patients were randomly assigned to the main study group (26 patients to FDI and 28 patients to the placebo) and constituted the intention-to-treat population. Follow-up was complete for all patients except two who missed their final follow-up visit, but all patients did not carry out all tests. The characteristics of the two arms of the study group were similar at baseline, although there was a 4 year age difference with the FDI group (*n* = 26; mean (SD) age 61.6 (10.1) years) vs. the placebo group (*n* = 28; mean (SD) age 57.8 (12.9) years). Mean (SD) serum creatinine was 167.0 (40.2) vs. 204.9 (67.3) micromol/L and eGFR 33.2 (9.3) vs. 29.1 (9.6) mL/min/1.73 m^2^ at baseline in FDI and placebo treated patients, respectively (Table 3).

### 3.2. Physical Function

At baseline, physical function scores were 36.5 (SD 19.6) in the placebo group, 38.9 (SD 5.9) in FDI, and 44.9 (SD 10.3) in Group 2. By 3 months, these changed to 37.5 (Placebo), 37.7 (FDI), and 41.3 (Group 2). This represents a 2.8% increase in placebo, a 2.9% decrease in FDI, and a 7.9% decline in Group 2. The difference between FDI and placebo for physical functioning was 5.8 percentage points. Repeated-measures ANOVA and paired *t*-tests found no statistically significant change (*p* = 0.786, placebo; *p* = 0.950, FDI) (Figure 1).

### 3.3. Role Physical

At baseline, prior to any intervention, mean role physical outcomes are poorer in those with CKD and iron deficiency (placebo and FDI) compared to those with CKD and no iron deficiency (Group 2). Baseline scores are as follows—placebo 42.8, FDI 42.1, Group 2 45.9. At 1 month placebo was 43.3, FDI was 42.3 and Group 2 was 43.8. At 3 months, scores were 37.5 for placebo, 40.9 for FDI, and 44.0 for Group 2. Following intervention at 3 months, the placebo group experienced a decrease of 12.28% in the mean score for role physical outcomes in comparison to their baseline mean score, whilst the FDI group only experienced a 3% decrease. As for Group 2, there was an overall decrease of 4.03% from baseline to 3 months (Figure 2). No statistically significant change was found (*p* = 0.211, placebo; *p* = 0.924, FDI).

### 3.4. Body Pain 

Body Pain scores were 42.3 (SD 19.6) in placebo, 40.8 (SD 11.6) in FDI, and 49.4 (SD 6.1) in Group 2 at baseline. At 1 month, placebo was 42.2, FDI was 43.9, and Group 2 was 47.5. At 3 months, placebo fell to 40.8 (−3.5%), FDI improved to 43.6 (+6.8%), and Group 2 declined to 47.4 (−4.0%). FDI therefore differed from placebo by 10.3 percentage points in outcome trajectory. Although there was a small decline in FDI at 1 month and 3 months, there was still an overall positive trajectory from baseline. There was a non-significant *p*-value (*p* = 0.65, placebo; *p* = 0.461, FDI) (Figure 3).

### 3.5. General Health

Baseline scores were 32.5 (SD 15.4) for placebo, 34.3 (SD 7.9) for FDI, and 42.1 (SD 5.8) for Group 2. At 1 month, scores were 32.1 for placebo, 38.3 for FDI, and 35.9 for Group 2. By 3 months, scores were 33.5 (placebo: +3.1%), 34.7 (FDI: +1.2%), and 40.2 (Group 2: −4.5%) (Figure 4). The difference in 3-month change between FDI and placebo was 1.9 percentage points. No statistically significant change was found (*p* = 0.825, placebo; *p* = 0.742, FDI).

### 3.6. Vitality

Vitality scores at baseline were 39.7 (SD 17.8) for placebo, 43.3 (SD 9.9) for FDI, and 48.1 (SD 10.3) for Group 2. At 1 month, scores were 42.5 for placebo, 44.0 for FDI, and 42.8 for Group 2. At 3 months: 41.4 (placebo: +4.2%), 44.0 (FDI: +1.8%), and 47.0 (Group 2: −2.3%) (Figure 5). The difference between placebo and FDI was 2.4 percentage points with placebo having a larger percentage increase at 3 months; however, *p*-values were not significant (*p* = 0.763, placebo; *p* = 0.465, FDI).

### 3.7. Social Functioning

Placebo started at 41.1 (SD 22.5), FDI at 41.3 (SD 10.7), and Group 2 at 42.0 (SD 10.9). At 1 month, placebo was 43.5, FDI was 43.5, and Group 2 was 46.9 (Figure 6). By 3 months, 41.8 (placebo: +1.7%), 43.0 (FDI: +4.1%), 43.4 (Group 2: +3.3%). FDI showed a 2.4 percentage point higher gain than placebo. Again, a non-significant *p*-values (*p* = 0.978, placebo; *p* = 0.433, FDI).

### 3.8. Role Emotional

Role emotional scores rose from 43.4 (SD 18.1) to 45.0 at 1 month and 47.5 at 3 months in placebo (+9.4%), 42.1 (SD 6.8) to 43.7 at 1 month and 45.4 at 3 months in FDI (+7.9%), and 49.4 (SD 16.1) to 49.0 at 1 month and 53.8 at 3 months in Group 2 (+8.9%) (Figure 7). All changes were statistically insignificant (*p* = 0.205, placebo; *p* = 0.771, FDI).

### 3.9. Mental Health

At baseline, the mean mental health score in the FDI group was 46.5 (SD 8.5), increasing to 50.2 at 1 month before settling at 49.9 at 3 months. In the placebo group, scores started at 45.0 (SD 19.6), improved slightly to 45.8 at 1 month, but declined to 42.3 at 3 months. Group 2 (CKD without iron deficiency) had a baseline score of 48.0 (SD 13.9), remained stable at 47.7 at 1 month, and reached 49.7 at 3 months (Figure 8). These changes were not statistically significant (*p* = 0.565, placebo; *p* = 0.167, FDI).

The difference in outcome trajectory between FDI and placebo from baseline to 3 months was 7.6 percentage points. Despite numerical improvements in the FDI group, repeated-measures ANOVA did not show statistically significant changes across time (linear *p* = 0.34). Likewise, independent samples *t*-tests revealed no significant difference between FDI and placebo groups at 3 months (*p* = 0.67).

### 3.10. Haemoglobin (Hb), Serum Ferritin (SF) and Transferrin Saturation (TSAT)

There was no statistically significant difference in Hb at 1 month (*p* = 0.195) and 3 months (*p* = 0.152) (Table 4). There was an upwards trend in Hb with an absolute mean increase of 1.7 g/L and 3.7 g/L with FDI compared to a fall of 1.0 g/L and 0.3 g/L in the placebo arm; thus, there was a difference of 2.7 g/L and 4.0 g/L in Hb in favour of FDI at 1 and 3 months, respectively. Overall, 96 and 95% of participants achieved an SF > 100 microgram/L at 1 and 3 months in the FDI group in comparison to 19 and 21% in the placebo group, respectively (*p* < 0.001, *p* < 0.001). Of the participants, 77 and 67% achieved a TSAT > 20% at 1 and 3 months in the FDI group in comparison to 23 and 30% in the placebo group, respectively (*p* < 0.001, *p* = 0.016).

## 4. Discussion

This study was designed to evaluate whether correcting iron deficiency before the onset of anaemia modifies patient-reported outcomes in non-dialysis-dependent CKD. Using data from the multicentre, double-blind Iron and the Heart Trial [9], we followed all eight domains of the Short Form-36 version 2 (SF-36v2) [10,14] for three months after a single 1000 mg infusion of FDI or placebo. The study therefore addresses a question that current nephrology guidelines acknowledge but do not resolve, whether iron deficiency per se, independent of haemoglobin concentration, should be treated in CKD to alleviate symptoms and improve quality of life.

Across the physical domains, role physical, physical function, bodily pain, and general health, scores declined in every group. However, the rate of decline was consistently lower in participants who received FDI. Role physical fell by only 3% in the FDI arm compared with 12% in placebo and 4% in iron-replete CKD controls. These results suggest that CKD is a progressive disease whether intervention is added or not. Both iron-deficient and non-iron-deficient groups experienced a decline in their role physical functioning from baseline to 3 months. However, we may be able to slow this progression by giving an intervention (FDI). There is a 9.3% difference in the role physical functioning between intervention groups, suggesting that giving FDI to correct iron deficiency may not improve role physical functioning entirely but may slow the deterioration of that outcome compared to if the iron deficiency is not corrected. Similar attenuation was observed for physical function and bodily pain. For physical function, FDI showed a numerically smaller decline than Group 2, suggesting a potential protective trend, albeit underpowered to detect significance. For bodily pain, the improvement in FDI compared to a decline in placebo and Group 2 suggests a potential treatment benefit to intervention. Additionally, FDI may stabilise general health more effectively than Group 2, which showed a progressive decline.

Mental health-related domains displayed an analogous pattern; the mental health score rose by 3.4 points in the FDI group but fell in placebo, creating a 7.6-point separation at three months. Although none of these differences reached statistical significance, and this was expected given that the study was not adequately powered, the uniform direction of change suggests a biologically coherent treatment effect rather than random fluctuation. The consistently positive direction of change in FDI suggests a mild but favourable trend.

The biological plausibility of these findings is supported by experimental and epidemiological evidence. Iron is indispensable for mitochondrial oxidative phosphorylation; deficiency impairs complex I activity, shifts energy production toward glycolysis, and reduces physical endurance in animal models without inducing anaemia [15]. Observational data in community-dwelling older adults further demonstrate independent associations between low iron indices and reduced hand-grip strength, a recognised marker of sarcopenia and frailty [16]. It is therefore reasonable to propose that timely intravenous iron repletion restores skeletal-muscle bioenergetics, delays fatigue, and slows the erosion of functional capacity that patients with CKD frequently report. The modest improvements observed in the psychosocial domains may reflect iron’s role as a co-factor for monoamine synthesis and its documented association with depressive symptoms [16].

Our results also echo the heart-failure literature, where three pivotal trials—FAIR-HF [7], CONFIRM-HF [17], AFFIRM-AHF [18], and IRONMAN [19]—all demonstrated that intravenous ferric carboxymaltose improved quality of life and reduced hospitalisation in patients with iron deficiency irrespective of haemoglobin status. The congruence between cardiac and renal populations—both characterised by chronic systemic inflammation, skeletal-muscle atrophy [20], and dysregulated iron metabolism—lends credence to the concept of treating iron deficiency as a stand-alone therapeutic target. Current KDIGO guidance recognises the high prevalence of iron deficiency in CKD but continues to anchor treatment recommendations to the presence of anaemia, citing insufficient evidence in non-anaemic cohorts [8]. The newly issued UK Kidney Association (UKKA) 2024/25 guideline provides UK-specific, graded recommendations that explicitly address iron deficiency *without* anaemia and endorse intravenous iron in appropriate people with non-dialysis-dependent CKD [21]. The present data therefore extend the evidence base by providing patient-centred outcome information from a rigorously blinded randomised study.

Clinically, these findings suggest that routine assessment of iron stores in stages G3b–G5 CKD could identify individuals likely to benefit from intravenous iron even when haemoglobin is within the conventional reference range. Oral preparations are often limited by poor gastrointestinal absorption and tolerance in CKD [22,23]; a single high-dose formulation that delivers full repletion in one visit offers practical advantages. Registry and trial data indicate that FDI is well tolerated, with a low incidence of hypersensitivity and a cardiovascular safety profile comparable to other intravenous preparations [17,23]. 

Our study did present with several limitations that would merit further research and careful consideration. First, the sample size was small (26 participants in the FDI arm), providing insufficient power to detect the minimal clinically important difference in each SF-36v2 domain. Second, follow-up was limited to three months; cardiology studies show the largest quality-of-life separation between four and twelve months, often after repeated dosing. This short follow-up period may therefore fail to capture the delayed responses in patients with concurrent co-morbidities that may be blunting early therapeutic effects. Extending follow-up in future trials would help determine both the trajectory and persistence of relief of symptoms over time. Third, incomplete questionnaire returns and attrition reduced the effective sample at later visits, and no imputation was undertaken. Fourth, potential confounding by baseline inflammation, comorbidity burden, or concurrent therapies was not adjusted for in this exploratory analysis. Finally, the cohort was recruited entirely from UK centres, which may restrict generalisability to populations with more pronounced inflammatory states or differing patterns of iron loss.

Future research should address these gaps through adequately powered, multicentre trials that include repeated high-dose intravenous iron to maintain ferritin levels and extend follow-up to at least twelve months and integrate both patient-reported and objective functional outcomes such as accelerometery, six-minute walk distance, and hand-grip dynamometry. Mechanistic sub-studies employing skeletal-muscle phosphorus magnetic-resonance spectroscopy or targeted metabolomic profiling would clarify the pathways through which iron repletion influences fatigue and physical performance. Comparative studies of ferric derisomaltose, ferric carboxymaltose, and iron isomaltoside are warranted to define the optimal balance of efficacy, tolerability, and cost. Parallel health-economic analyses will be essential to determine whether attenuating QoL decline translates into delayed dialysis initiation or reduced hospital utilisation.

## 5. Conclusions

In conclusion, a single 1000 mg infusion of ferric derisomaltose did not yield statistically significant improvements in SF-36v2 scores over three months but consistently attenuated deterioration in both physical and mental domains relative to placebo. These findings, supported by mechanistic and cross-disciplinary evidence, indicate that iron deficiency itself is a modifiable contributor to symptom burden in non-dialysis-dependent CKD. Confirmation in larger, longer-duration trials is required, but the data presented here support a paradigm in which proactive identification and correction of iron deficiency become integral components of comprehensive CKD care.

## Figures and Tables

**Figure 1 biomedicines-13-01860-f001:**
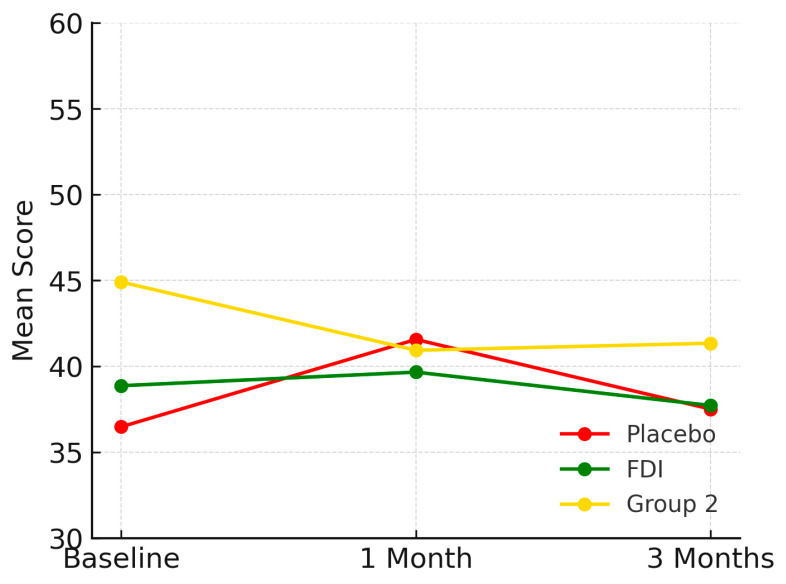
Demonstrates physical function domain for each patient group and their mean scores across the study period (baseline, 1 month, 3 months). While all groups show some variability, FDI demonstrates a relatively stable trajectory, with less fluctuation than placebo and a slower decline than Group 2. Placebo = red, FDI (ferric derisomaltose) = green, and Group 2 (CKD without iron deficiency) = yellow.

**Figure 2 biomedicines-13-01860-f002:**
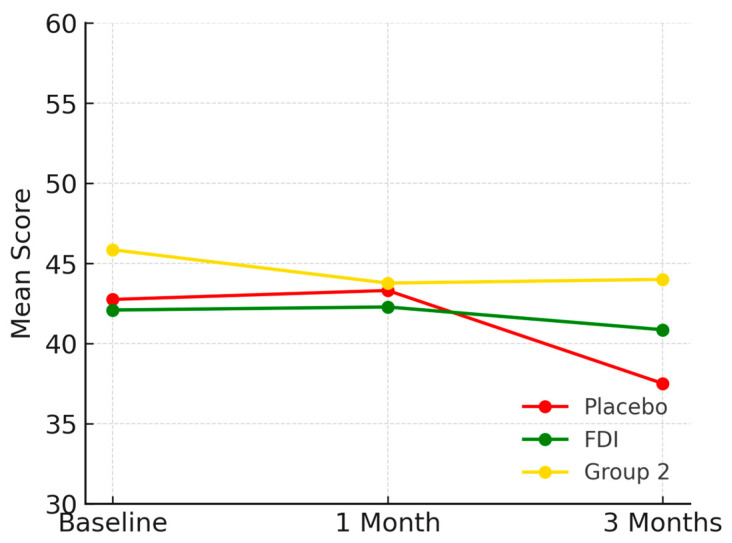
Depicts the role physical domain over time. Mean scores declined most in the placebo group. Placebo = red, FDI (ferric derisomaltose) = green, and Group 2 (CKD without iron deficiency) = yellow.

**Figure 3 biomedicines-13-01860-f003:**
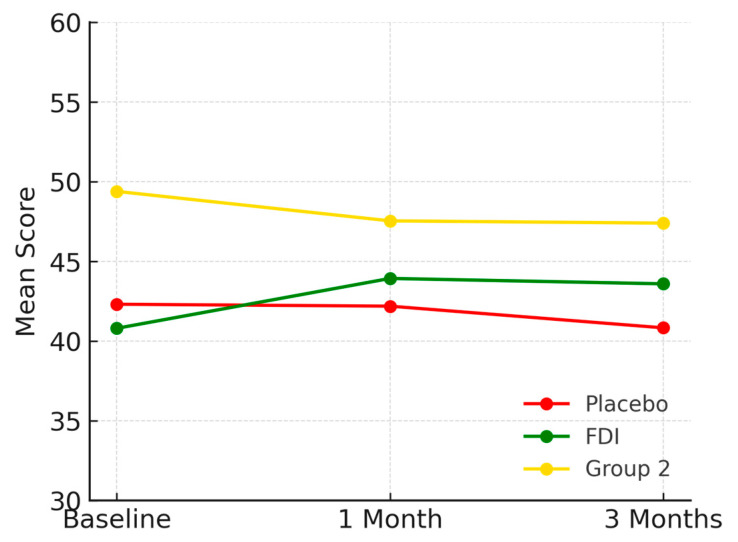
Illustrates Body Pain scores over the study duration. The FDI group showed a consistent upward trend, in contrast to the placebo and Group 2 groups, which declined slightly. Placebo = red, FDI (ferric derisomaltose) = green, and Group 2 (CKD without iron deficiency) = yellow.

**Figure 4 biomedicines-13-01860-f004:**
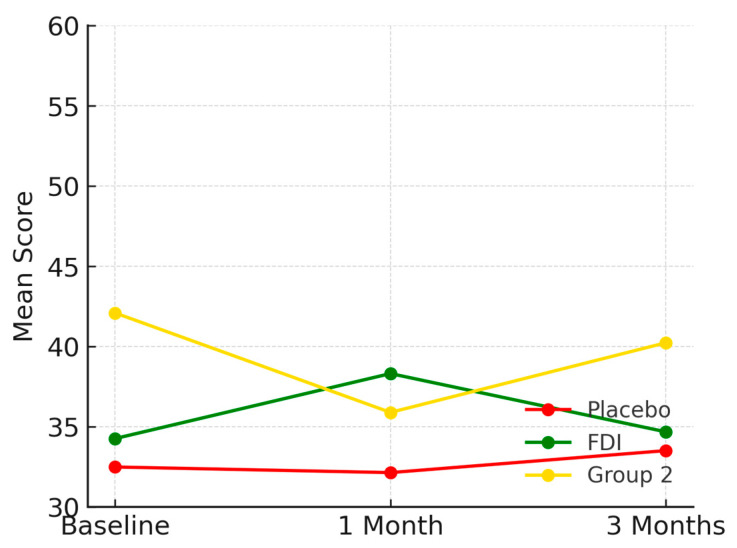
Shows the general health domain. FDI showed a short-lived peak at 1 month before returning close to baseline, while the placebo and Group 2 groups demonstrated minimal or declining trajectories. Placebo = red, FDI (ferric derisomaltose) = green, and Group 2 (CKD without iron deficiency) = yellow.

**Figure 5 biomedicines-13-01860-f005:**
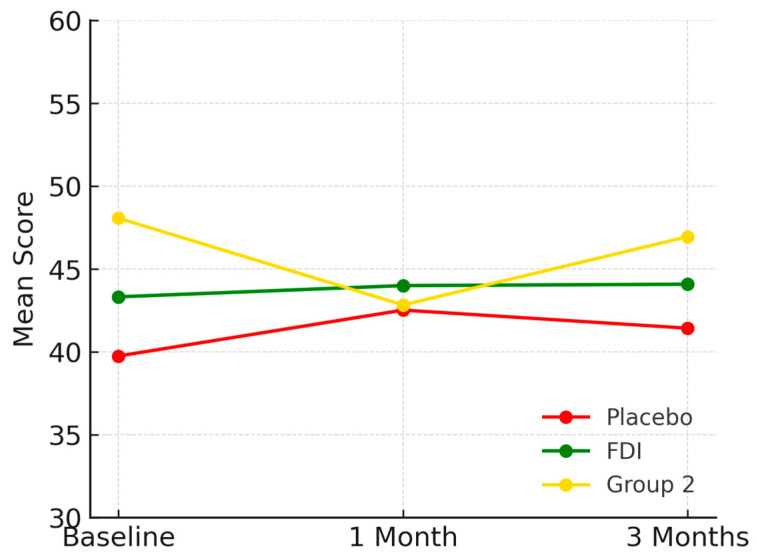
Visualises vitality domain scores. The FDI group exhibited a sustained, slight improvement. Placebo showed initial gain followed by decline, while Group 2 steadily decreased then partially recovered. Placebo = red, FDI (ferric derisomaltose) = green, and Group 2 (CKD without iron deficiency) = yellow.

**Figure 6 biomedicines-13-01860-f006:**
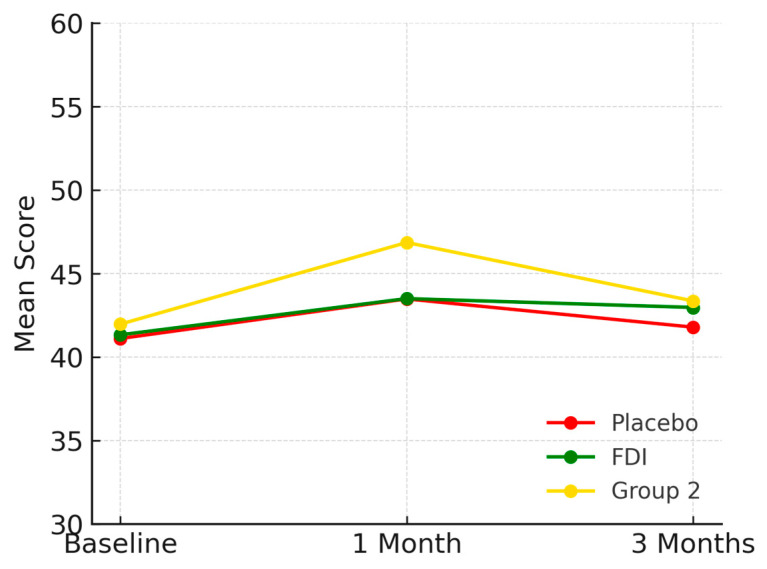
Plots social functioning across timepoints. All groups showed modest improvement, though the FDI group maintained slightly higher scores at 3 months, suggesting a subtle but consistent benefit. Placebo = red, FDI (ferric derisomaltose) = green, and Group 2 (CKD without iron deficiency) = yellow.

**Figure 7 biomedicines-13-01860-f007:**
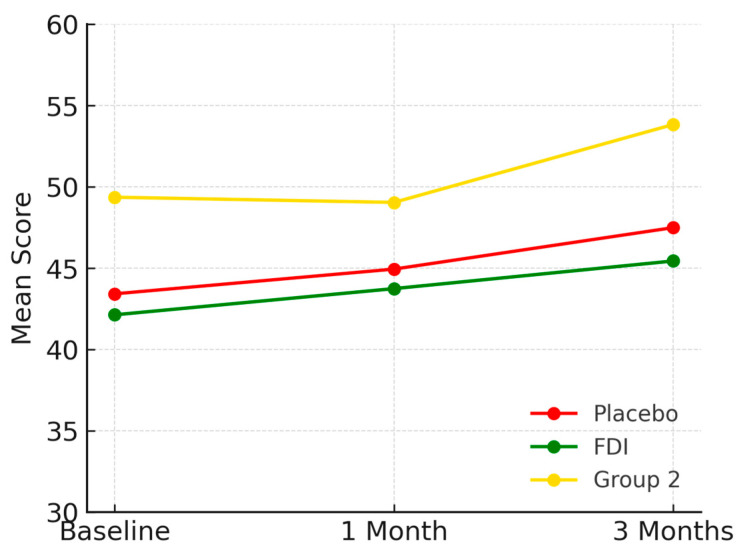
Displays role emotional outcomes. All groups improved over time, particularly Group 2. Placebo = red, FDI (ferric derisomaltose) = green, and Group 2 (CKD without iron deficiency) = yellow.

**Figure 8 biomedicines-13-01860-f008:**
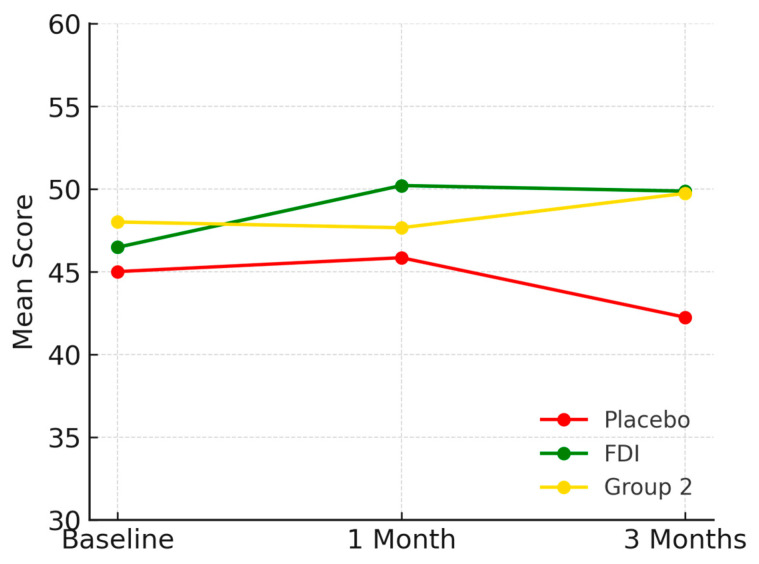
Represents mental health scores. The FDI group showed a marked increase sustained through 3 months. Placebo scores declined. Group 2 remained relatively stable. Placebo = red, FDI (ferric derisomaltose) = green, and Group 2 (CKD without iron deficiency) = yellow.

**Table 1 biomedicines-13-01860-t001:** Full inclusion and exclusion criteria for the Iron and Heart Study [9].

Inclusion and Exclusion Criteria of Iron and Heart Study	
Inclusion	Exclusion
Patients with established CKD (Stages 3b-5) not on dialysis	Pregnancy or breast feeding
Resting BP ≤ 160/90 mmHg	Weight ≤ 50 kg
Aged 18–80 years	Known allergy to iron therapy
Serum ferritin < 100 μg/L; AND/OR transferrin saturation ≤ 20%	Haemochromatosis or history of acquired iron overload
Haemoglobin ≥ 110 g/L AND ≤150 g/L	Parenteral iron received within previous 6 weeks
	CRP > 50 mg/L
	Active infection
	Current therapy with ESA agents
	Patients with atrial fibrillation
	Patients with solid organ cancer
	Patients with known haemoglobinopathy, myelodysplasia or myeloma
	Patients with musculoskeletal disease, who the investigator deems unable to carry out the 6-min walking test.

**Table 2 biomedicines-13-01860-t002:** Mapping of SF-36 domains within the KDQoL instrument: item numbers and question-level breakdown.

Mapping of SF-36 Domains				
Domain	Number of Items	Item Numbers	Description	All Questions
Physical Function	10	Q3a–Q3j	Assesses limitations in physical activities such as walking, climbing stairs, and carrying groceries (Q3a–Q3j).	Q3a–Q3j: Moderate activities, climbing stairs, bending/kneeling, walking various distances, bathing/dressing, lifting groceries.
Role Physical	4	Q4a–Q4d	Evaluates problems with work or daily activities due to physical health (Q4a–Q4d).	Q4a–Q4d: Cut down time spent on work, accomplished less, limited in kind of work, difficulty performing work due to physical health.
Body Pain	2	Q7, Q8	Measures pain intensity and interference with normal work (Q7, Q8).	Q7, Q8: Pain intensity in past 4 weeks; pain interference with normal work including both outside the home and housework.
General Health	5	Q1, Q11a–Q11d	Captures general perceptions of health, including current state and future expectations (Q1, Q11a–Q11d).	Q1, Q11a–Q11d: Self-rated general health, health is excellent, health is likely to get worse, health compared to others, getting sick easier than others.
Vitality	4	Q9a, Q9e, Q9g, Q9i	Assesses energy and fatigue levels over recent time (Q9a, Q9e, Q9g, Q9i).	Q9a, Q9e, Q9g, Q9i: Felt full of life, had a lot of energy, felt tired, felt worn out.
Social Functioning	2	Q6, Q10	Evaluates interference of health problems with social activities (Q6, Q10).	Q6, Q10: Interference with social activities due to physical or emotional problems; time spent on social activities.
Role Emotional	3	Q5a–Q5c	Measures difficulties in role functioning due to emotional problems (Q5a–Q5c).	Q5a–Q5c: Cut down amount of time on work, accomplished less, did work less carefully due to emotional problems.
Mental Health	5	Q9b, Q9c, Q9d, Q9f, Q9h	Assesses emotional well-being and psychological distress (Q9b, Q9c, Q9d, Q9f, Q9h).	Q9b, Q9c, Q9d, Q9f, Q9h: Felt nervous, felt so down nothing could cheer you up, felt calm and peaceful, felt downhearted and blue, felt happy.

**Table 3 biomedicines-13-01860-t003:** Participants with chronic kidney disease (CKD) and iron deficiency without anaemia (Group 1), and sub-grouped by randomisation arm. Data are presented as mean (SD), number (*n*), or percentage (%) unless otherwise stated.

Baseline Characteristics of Participants				
	Total (*n* = 54)	FDI (*n* = 26)	Placebo (*n* = 28)	*p*-Value
Age (years)	59.6 (11.7)	61.6 (10.1)	57.8 (12.9)	>0.05
Range	32–78	37–78	32–78	
Sex, *n* (%)				
Male	26 (49%)	11 (42.3%)	15 (53.6%)	>0.05
Female	27 (51%)	14 (53.8%)	13 (46.4%)	>0.05
Unknown	1	1	0	
Ethnicity, *n* (%)				
White	42 (78%)	19 (73.1%)	23 (82.1%)	>0.05
Asian	3 (5%)	1 (3.8%)	2 (7.1%)	>0.05
Black	4 (7%)	4 (15.4%)	3 (10.7%)	>0.05
Mixed Race	1 (2%)	1 (3.8%)	0 (0%)	>0.05
Unknown/Other	1 (2%)	1 (3.8%)	0 (0%)	>0.05
Smoker, *n* (%)				
Current	5 (9%)	1 (3.8%)	4 (14.3%)	>0.05
Previous	17 (32%)	8 (30.8%)	9 (32.1%)	>0.05
No	32 (59%)	17 (65.4%)	15 (53.6%)	>0.05
BMI (kg/m^2^)	30.3 (6.5), *n* = 53	30.7 (6.8)	30.0 (6.4)	>0.05
Serum Ferritin (µg/L)	66.3 (44.1), *n* = 53	64.2 (29.1)	68.4 (55.3)	0.195
TSAT (%)	20.1 (7.4), *n* = 53	22.3 (8.8)	19.7 (5.6)	<0.001
Haemoglobin (g/L)	128.7 (10.1), *n* = 54	131 (7.4), *n* = 26	126.6 (11.8), *n* = 28	<0.001
Serum Creatinine (µmol/L)	186.7 (58.6), *n* = 54	167 (40.2), *n* = 26	204.9 (67.3), *n* = 28	<0.001
eGFR (mL/min/1.73 m^2^)	31.1 (9.6), *n* = 54	33.2 (9.3), *n* = 26	29.1 (9.6), *n* = 28	>0.05
Cystatin C (mg/L)	2.2 (0.6), *n* = 52	2.1 (0.5), *n* = 26	2.4 (0.6), *n* = 26	>0.05
uACR (mg/mmol)	60.9 (133.3), *n* = 26	26.9 (40.1), *n* = 13	94.8 (181.4), *n* = 13	>0.05
uPCR (mg/mmol)	83.8 (128.4), *n* = 40	51.9 (59.3), *n* = 19	112.7 (164.8), *n* = 21	>0.05
CRP (mg/L)	5.0 (4.4), *n* = 53	6.3 (5.5), *n* = 26	3.8 (2.4), *n* = 27	>0.05
Serum Albumin (g/L)	39.8 (5.6)	40.8 (4.2)	38.8 (6.9)	>0.05
Platelet Count (×10^9^/L)	235 (58.9)	226 (52.4)	243 (64.1)	>0.05
Phosphate (mmol/L)	1.2 (0.2)	1.1 (0.2)	1.2 (0.2)	>0.05
NT pro BNP (ng/L)	4852.0 (12,684.4), *n* = 51	4225.0 (8819.5), *n* = 25	5454.0 (15,695.0), *n* = 26	>0.05
PWV (m/s)	8.3 (3.2), *n* = 54	8.3 (2.8), *n* = 26	8.3 (3.6), *n* = 28	>0.05
AiX (%)	24.2 (10.7), *n* = 54	25.4 (10.7), *n* = 26	24.1 (10.8), *n* = 28	>0.05

**Table 4 biomedicines-13-01860-t004:** Values shown are mean (SD), *p* values derived from independent samples *t*-tests comparing FDI and placebo at each timepoint.

Change in Haemoglobin, Ferritin, and Transferrin Saturation over Time				
	Timepoint	FDI (*n* = 26)	Placebo (*n* = 28)	*p* Value
Haemoglobin (g/L)	Baseline	131.0 (7.4)	126.5 (11.8)	—
	1 month	132.7 (7.2)	125.5 (12.8)	0.195
	3 months	134.7 (8.9)	126.2 (12.4)	0.152
Serum Ferritin (µg/L)	Baseline	64.2 (29.1)	68.4 (55.3)	—
	1 month	266.0 (105.8)	70.5 (55.8)	<0.001
	3 months	234.4 (105.3)	69.1 (59.8)	<0.001
TSAT (%)	Baseline	22.3 (8.8)	19.7 (5.6)	—
	1 month	29.6 (9.5)	19.1 (7.1)	<0.001
	3 months	26.4 (10.5), *n* = 21	18.0 (6.8), *n* = 23	<0.001

## Data Availability

The original contributions presented in this study are included in the article. Further inquiries can be directed to the corresponding author.

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
