# Peer review of "Investigating the Impact of Ferric Derisomaltose (FDI) on Patient-Reported Quality-of-Life Outcome Measures in Iron-Deficient but Not Anaemic Patients with Chronic Kidney Disease"

_biomedicines, 2025, doi:10.3390/biomedicines13081860_

Round 1
Reviewer 1 Report
Comments and Suggestions for Authors
Authors of this study evaluated the impact of a single 1000 mg dose of ferric deriso-maltose (FDI) on patient-reported QoL outcomes in CKD stages G3b–G5. They included 54 participants with iron deficiency without anaemia (IDNA) that were randomised 1:1 to receive either FDI or placebo, with iron-replete patients with CKD as controls. For this purpose, SF-36v2 surveys were completed at baseline, 1 month, and 3 months. They found out that scores declined in all groups regarding physical domains but deterioration was consistently smaller in the FDI group. None of the differences reached statistical significance, however, the consistent trend across all domains favoured FDI, suggesting that IV iron may attenuate QoL decline in patients with non-dialysis dependent CKD with IDNA.
It is an interesting study with some value for clinical practice as well, especially regarding the frequency of CKD and associated complications such as iron deficiency with or without anaemia. The study on participants with iron deficiency without anaemia is a new and original aspect targeting earlier stages of CKD with already present symptoms due to this complication. In general, the introduction provides sufficient background and the results support conclusions. However, there are some concerns regarding this study that should be addressed, such as:
1. Abstract:
"Mental Health scores increased by 3.4 points in the FDI group but declined by 2.7 points in the placebo group, creating a 7.6-point separation."
- Comment: the sum of 3.4 + 2.7 is 6.1 points. Authors should explain how they have got a 7.6-point separation.
2. METHODS, 1st paragraph:
The Iron and the Heart Trial ...................... compared to a placebo infusion at baseline (0 months), and assessed at 1 month and 3-months post-infusion, to evaluate the impact of FDI on exercise capacity.
- Comment: the follow-up period is very short. However, it is true that authors commented this concern later on in the article, stating "Future research should address these gaps through adequately powered, multicentre trials that include repeated high-dose intravenous iron to maintain ferritin levels and extend follow-up to at least twelve months." Nevertheless, this is a significant limitation of the study and authors should either provide some more data about long-term outcome measures, if available, or describe in more detail why the period of three months could be justified, supported with more evidence from the literature.
3. METHODS, 2nd paragraph:
54 participants with iron deficiency but no anaemia (Group 1) were randomly assigned to either FDI or placebo (1:1 ratio)...........................
- Comment: this is a small sample size, especially regarding the frequency of CKD in adult population. However, the authors commented this concern later on in the article, stating "Future research should address these gaps through adequately powered, multicentre trials that include repeated high-dose intravenous iron to maintain ferritin levels and extend follow-up to at least twelve months." as well as in Conclusions: "............Confirmation in larger, longer-duration trials is required,....................." Nevertheless, this is a significant limitation of the study and authors should comment in more detail about reasons for choosing this sample size, and the consequences for the validy of results.
4. METHODS
Table 1: The questionnaire should be presented in more detail, prefferably by adding the KDQoL-SF-36 survey as an attachment or supplementary file. In addition, a mechanism for alocating a certain number of points to certain items must be described.
5. RESULTS
3.2. Role Physical, 3.6. Social Functioning3.8. Mental Health: There are no statistical results presented, including p-value.
3.4. General Health: Results are presented at baseline and at 3 months but not at 1 month. However, fig. 4 presents data also at 1 month. Please explain.
6. RESULTS, 1st paragraph:
"The difference in outcome trajectory between FDI and Placebo from baseline to 3 months was 7.6 percentage points. Despite numerical improvements in the FDI group, .......................................... revealed no significant difference between FDI and placebo groups at 3 months (p = 0.67). - Does this reffer to a special domain or to results in general? Please, explain.
7. DISCUSSION, 2nd paragraph:
Across the physical domains, Role Physical, Physical Function, Bodily Pain and Gen-eral Health, scores declined in every group, consistent with the progressive nature of advanced CKD.
- Comment: 3 months is a period too short to be explained by the progressive nature of advanced CKD. Authors should explain whether other parameters (such as glom. filtration rate, hyperparathyroidism etc.) also declined to such extent in these patients.
8. DISCUSSION, 3rd paragraph:
"Mental-health-related domains displayed an analogous pattern. Non-significant p-values for social functioning (p = 0.43 FDI) do not rule out clinical benefit."
- Comment: this is a speculation that was not objectively confirmed in this study. This sentence should, therefore, be either deleted or supported by more evidence from the literature and cited by relevant references.
Reviewer 2 Report
Comments and Suggestions for Authors
All comments and revisions have been provided within the manuscript.

Author Response
Thank you for taking the time to review our manuscript. We appreciate your constructive feedback and thoughtful comments. Please find our detailed responses to each point in the attached PDF document.

Round 2
Reviewer 2 Report
Comments and Suggestions for Authors
Almost all of my comments/revisions have been applied by authors.
The only remained comment is adding p values related to statistical analyses of variables between FDI & Placebo groups in Table 3.
Author Response
Please note that all changes have been made and table 3 now contains the p-values.